# Mesenchymal Stem/Stromal Cells Induce Myeloid-Derived Suppressor Cells in the Bone Marrow via the Activation of the c-Jun N-Terminal Kinase Signaling Pathway

**DOI:** 10.3390/ijms25021119

**Published:** 2024-01-17

**Authors:** Hyun Ju Lee, Joo Youn Oh

**Affiliations:** 1Laboratory of Ocular Regenerative Medicine and Immunology, Biomedical Research Institute, Seoul National University Hospital, 101 Daehak-ro, Jongno-gu, Seoul 03080, Republic of Korea; dalmuly@empas.com; 2Department of Ophthalmology, Seoul National University College of Medicine, 103 Daehak-ro, Jongno-gu, Seoul 03080, Republic of Korea

**Keywords:** bone marrow, JNK, MAPK, mesenchymal stem/stromal cells, myeloid-derived suppressor cells, signaling pathway

## Abstract

Our previous study demonstrated that mesenchymal stem/stromal cells (MSCs) induce the differentiation of myeloid-derived suppressor cells (MDSCs) in the bone marrow (BM) under inflammatory conditions. In this study, we aimed to investigate the signaling pathway involved. RNA-seq revealed that the mitogen-activated protein kinase (MAPK) pathway exhibited the highest number of upregulated genes in MSC-induced MDSCs. Western blot analysis confirmed the strong phosphorylation of c-Jun N-terminal kinase (JNK) in BM cells cocultured with MSCs under granulocyte-macrophage colony-stimulating factor stimulation, whereas p38 kinase activation remained unchanged in MSC-cocultured BM cells. JNK inhibition by SP600125 abolished the expression of *Arg1* and *Nos2*, hallmark genes of MDSCs, as well as *Hif1a*, a molecule mediating monocyte functional reprogramming toward a suppressive phenotype, in MSC-cocultured BM cells. JNK inhibition also abrogated the effects of MSCs on the production of TGF-β1, TGF-β2 and IL-10 in BM cells. Furthermore, JNK inhibition increased *Tnfa* expression, while suppressing IL-10 production, in MSC-cocultured BM cells in response to lipopolysaccharides. Collectively, our results suggest that MSCs induce MDSC differentiation and promote immunoregulatory cytokine production in BM cells during inflammation, at least in part, through the activation of the JNK–MAPK signaling pathway.

## 1. Introduction

Mesenchymal stem/stromal cells (MSCs) maintain tissue homeostasis by regulating immune responses and promoting tissue regeneration. The depletion of monocytes or macrophages abrogates the immunomodulatory and regenerative activities of MSCs in multiple disease models, pointing to the crucial role of innate immune cells as essential mediators of MSCs’ homeostatic action [1,2]. A number of studies have elucidated the actions and mechanisms of MSCs, revealing their capacity to suppress pro-inflammatory immune cells and induce immunosuppressive cells. Notably, among the suppressive immune cells induced by MSCs are myeloid-derived suppressor cells (MDSCs).

Many studies by our and other groups have demonstrated that MSCs induce the differentiation, expansion and recruitment of MDSCs. For instance, Chen et al. observed that conditioned media from human umbilical cord (UC)-derived MSCs skew the development of mouse monocyte-derived dendritic cells toward an MDSC phenotype exhibiting CD8^+^ T cell-suppressive activity [3]. Yen et al. reported that human bone marrow (BM)-derived MSCs enhance the expansion of MDSCs in human peripheral blood leukocytes, which inducing regulatory T cells [4]. Also, our group previously demonstrated that the intravenous infusion of human BM MSCs ameliorates experimental autoimmune uveoretinitis (EAU) in mice by recruiting monocytic MDSCs to sites of inflammation [5]. Similarly, Wang et al. showed that human UC MSCs alleviate graft-versus-host disease (GVHD) by enriching MDSCs in GVHD target tissues [6]. More recently, the Le Blanc group reported that human BM MSCs increase the levels of both mononuclear and polymorphonuclear MDSCs in human whole blood [7]. These findings collectively underscore the involvement of MSDCs in mediating the immunoregulatory action of MSCs.

MDSCs constitute a heterogeneous group of myeloid cells with potent immunosuppressive activity. Recent data uncovered that the development of MDSCs occurs in two partially overlapping phases [8,9]: One is the expansion and conditioning of myeloid cells in the BM and spleen, and the other is the conversion of neutrophils and monocytes into pathologically activated MDSCs in peripheral tissues. MDSCs accumulate in tissues in various pathologic conditions, including cancer, infection, acute and chronic inflammation and autoimmunity, and they play an important role in regulating immune responses. The immunoregulatory activity of MDSCs can be detrimental or beneficial [8,9]. For instance, in cancer, MDSCs exacerbate the disease and are generally associated with poor clinical outcomes. In contrast, in autoimmune and inflammatory disorders, the immunosuppressive functions of MDSCs contribute to limiting disease severity. 

Although emerging data have elucidated the genomic, proteomic and metabolic characteristics of MDSCs, the precise mechanisms that drive neutrophils and monocytes to differentiate into MDSCs remain unclear. Several molecular pathways that regulate the suppressive functions of MDSCs have been suggested, but they largely vary depending on the factors and tissue microenvironments inducing MDSC development [10,11,12]. Identifying the specific pathways leading to MDSC differentiation and activation would provide valuable information for the potential clinical targeting of MDSCs for the treatment of various diseases. 

In a previous study, our group demonstrated that human BM-derived MSCs direct the differentiation of BM cells from pro-inflammatory CD11b^hi^Ly6C^hi^Ly6G^lo^ monocytes to CD11b^mid^Ly6C^mid^Ly6G^lo^ MDSCs in vitro under granulocyte-macrophage colony-stimulating factor (GM-CSF) stimulation and in mice with EAU [13]. In the present study, we aimed to explore the signaling pathway in BM cells through which MSCs induce MDSC differentiation and regulate their suppressive functions.

## 2. Results 

### 2.1. MAPK-Related Genes Are Upregulated in MSC-Induced MDSCs 

In our prior study [13], MSCs were found to induce the expression of *Arg1* and *Nos2*, hallmark MDSC genes encoding the immunosuppressive enzymes arginase and inducible nitric oxide synthase (iNOS), respectively [8,9], in GM-CSF-stimulated BM cells. Also, MSCs elevated the production of immunosuppressive cytokines, such as IL-10, TGF-β1 and TGF-β2, in GM-CSF-stimulated BM cells, while suppressing TNF-α secretion [13]. Remarkably, the MSC-induced MDSCs exhibited a CD11b^mid^Ly6C^mid^Ly6G^lo^ phenotype, distinct from the pro-inflammatory CD11b^hi^Ly6C^hi^Ly6G^lo^ monocytes differentiated from GM-CSF-stimulated BM cells [13]. 

Building upon these findings, we conducted a screening for the signaling pathway involved in MSC-induced MDSC differentiation. To this end, we analyzed RNA-seq data from CD11b^mid^Ly6C^mid^Ly6G^lo^ MDSCs derived from MSC-cocultured BM cells under GM-CSF stimulation and compared them with those of CD11b^hi^Ly6C^hi^Ly6G^lo^ monocytes sorted from GM-CSF-stimulated BM cells without MSC coculture (ArrayExpress accession E-MTAB-8975) (Figure 1A).

A comparative Kyoto Encyclopedia of Genes and Genomes (KEGG) pathway analysis using the DAVID tool [14,15] revealed that the mitogen-activated protein kinase (MAPK) pathway, along with the phosphatidylinositol 3-kinase (PI3K)/Akt pathway, exhibited the highest numbers of upregulated genes in MSC-induced MDSCs compared to BM cells without MSC coculture. Since our group had previously elucidated the effects of MSCs on the Akt/mTOR complex 1 pathway in monocytes/macrophages [16], we chose to focus on the MAPK pathway for further investigation. The identified upregulated genes related to the MAPK pathways are listed in Table 1, and the pathways in which they are involved are illustrated in Figure 1B. 

In contrast, pathway analysis using the KEGG database revealed that multiple inflammation-related pathways, such as NOD-like receptor, NF-κB, chemokine, TNF and toll-like receptor signaling pathways, were included in the biological pathways involving downregulated genes in MSC-induced MDSCs compared to BM cells without MSC coculture. Notably, the NOD-like receptor signaling pathway included the highest number of downregulated genes in MSC-induced MDSCs (Table 2).

### 2.2. MSCs Activate the JNK Pathway in BM Cells

Based on the observed upregulation of MAPK-related genes in MSC-induced MDSCs (Figure 1B, Table 1), we then investigated the activation of c-Jun N-terminal kinase (JNK) and p38 MAPK signaling pathways, members of the MAPK families [17], in BM cells with and without MSC coculture in the presence or absence of GM-CSF. Western blot analysis showed that GM-CSF alone did not significantly influence JNK or p38 phosphorylation in BM cells. Remarkably, MSC coculture induced the robust phosphorylation of JNK in GM-CSF-stimulated BM cells (Figure 2A,B). In contrast, while JNK activation was prominent, p38 phosphorylation showed no significant increase in BM cells via MSC coculture (Figure 2A,B). 

These findings indicate that MSCs specifically activate the JNK–MAPK signaling pathway in GM-CSF-stimulated BM cells. 

### 2.3. JNK Inhibition Abrogates MSC Effects on MDSC Induction

To assess the specific contribution of JNK in the MSC-induced differentiation of BM cells, we utilized the anthrapyrazolone inhibitor SP600125 as a JNK inhibitor [18]. Initially, we confirmed the efficacy of SP600125 in inhibiting JNK phosphorylation in a dose-dependent manner. Treatment of BM cells with 40 μM SP600125 for 3 d negated the effect of MSCs on JNK activation in BM cells (Figure 3A). 

Subsequently, we examined the impact of JNK inhibition on the expression of molecules mediating the immunosuppressive functions of MDSCs in BM cells cocultured with MSCs (Figure 3B). The results revealed that MSCs highly induced the expression of *Arg1*, *Nos2* and *Hif1a* in BM cells (Figure 3C). *Arg1* and *Nos2* are hallmark MDSC genes encoding immunosuppressive enzymes, arginase and iNOS, respectively [8,9]. *Hif1a* encodes hypoxia-inducible factor (HIF)-1α, which has been shown to mediate the functional reprogramming of monocytes into a suppressive phenotype [19]. Interestingly, JNK inhibition by SP600125 significantly downregulated the expression of *Arg1, Nos2* and *Hif1a* (Figure 3C). Similar results were observed with the production of immunoregulatory cytokines. BM cells cocultured with MSCs exhibited elevated levels of immunosuppressive cytokines TGF-β1 and TGF-β2 compared to those without MSC coculture (Figure 3D). However, JNK inhibition by SP600125 significantly reduced the secretion of TGF-β1 and TGF-β2 in BM cells cocultured with MSCs (Figure 3D). 

Furthermore, we investigated the response of BM cells to lipopolysaccharide (LPS) stimulation (Figure 3B). As expected, LPS treatment markedly upregulated the expression of pro-inflammatory cytokines TNF-α and IL-12 in BM cells (Figure 3E). The MSCs almost completely suppressed the upregulation of TNF-α and IL-12 in BM cells in response to the LPS (Figure 3E). SP600125 treatment significantly reversed the effects of the MSCs on TNF-α suppression in BM cells, while the secretion of IL-12 remained unaffected by SP600125 (Figure 3E). In addition, MSCs induced IL-10 secretion in BM cells in response to LPS stimulation, for which its effect was significantly attenuated by SP600125 (Figure 3E).

Collectively, these results demonstrate that MSCs induce MDSC genes and immunosuppressive molecules in BM cells under GM-CSF stimulation and mitigate their pro-inflammatory activation in response to the LPS. The inhibition of the JNK pathway reverses these effects of MSCs on BM cells. 

## 3. Discussion 

Our results demonstrated that a significant number of genes related to MAPK pathways were upregulated in MDSCs induced by MSCs in the BM. Specifically, the JNK pathway shows prominent activation in BM cells cocultured with MSCs. The blockade of JNK activation in BM cells abolished the effects of MSCs on the induction of hallmark MDSC genes and immunosuppressive molecules mediating the MDSC function. Also, JNK inhibition abrogated the suppressive action of MSCs on the inflammatory activation of BM cells in response to LPS. Therefore, the results suggest that the JNK–MAPK signaling pathway is crucial for MSCs’ actions to promote immunosuppressive molecules and induce MDSC differentiation in BM cells during inflammation. 

The JNK signaling pathway is a member of the MAPK signal transduction pathways, which relay, amplify and integrate signals from a range of extracellular stimuli and thereby elicit multiple physiological processes in mammalian cells, including cell proliferation, differentiation, apoptosis and inflammatory responses [17]. The JNK pathway has been implicated in diverse innate immune responses. For example, JNK inhibition was reported to decrease IL-10 expression in stimulated human monocytes [20]. Also, JNK inhibition significantly reduces the Pam3CSK4 (PAM3)-dependent generation of immunosuppressive macrophages in human monocytes [21]. Moreover, JNK inhibition was shown to increase the apoptosis of both granulocytic and monocytic MDSCs [22]. In our study, MSCs induced JNK activation in BM cells, leading to the induction of immunoregulatory molecules (*Arg1*, *Nos2*, *Hif1α*, TGF-β1, TGF-β2 and IL-10) that mediate the immunosuppressive function of MDSCs. These findings indicate that JNK activation contributes to the action, survival and differentiation of MDSCs and immunosuppressive macrophages. Given the highly heterogeneous and context-dependent nature of MDSCs and monocytes/macrophages [12], it can be suggested that other signaling pathways, in addition to the JNK pathway, are also involved in MDSC differentiation and functions. In our study, JNK inhibition did not alter the effect of MSCs on IL-12 suppression in BM cells, while it significantly abrogated the effect on TNF-α and IL-10 levels, pointing to the involvement of other pathways in MSC-mediated IL-12 downregulation in BM cells. In a study by Bayik et al., JNK contributed to the PAM3-driven generation of suppressive macrophages, but it was not involved in M-CSF-dependent suppressive macrophage polarization [21]. Together, these findings raise the possibility that multiple pathways might be implicated in the differentiation of MDSCs and immunosuppressive macrophages. Considering that various pathways, including PI3K, Ras, Jak/Stat and TGF-β pathways, have been demonstrated to play a role in the generation of MDSCs from myeloid precursors in various disease conditions [10], it is plausible that other pathways, along the JNK pathway, might contribute to MSC-induced MDSC differentiation. Indeed, our KEGG pathway analysis revealed that several cell proliferation/differentiation-related signaling pathways, such as PI3K-Akt, Wnt and Hippo signaling pathways, were also included in the pathways exhibiting the upregulated genes in MSC-induced MDSCs. Future studies will be required to investigate the roles of these pathways in MDSC differentiation induced by MSCs.

Conflicting results on the role of JNK in innate immune responses have also been reported. A study by Deng et al. demonstrated that JNK activation by synthetic bacterial lipoprotein, a TLR1/2 agonist, redirected the differentiation of monocytic MDSC toward M1 pro-inflammatory macrophages [23]. Also, a study by Liu et al. reported that TNF-α-stimulated gene/protein 6 secreted by human umbilical cord-derived MSCs inhibited the activation of both p38 and JNK–MAPK pathways in peritoneal macrophages and thus attenuated inflammation in a skin wound after a burn injury [24]. These discrepancies suggest that the signaling pathways involved in the polarization of monocytes/macrophages and MDSC differentiation depend on the cell type and extracellular stimuli [21,25] and that JNK activation can be pro-inflammatory or immunosuppressive according to cell types and tissue microenvironments. As such, the identification of specific signals and pathways that direct MDSC differentiation and lead to suppressive modes in myeloid populations in different disease settings would be important for improving the clinical outcomes of MDSC-based therapies. 

The exogenous administration of MSCs has proven beneficial in ameliorating acute inflammatory pathology and autoimmune diseases, making MSC-based therapies the subject of extensive clinical trials [26]. One of the key mechanisms underlying the therapeutic effects of MSCs is their ability to generate suppressive macrophages or MDSCs. However, the immunosuppressive environment created by MSCs raises concerns about potential tumor promotion in hosts receiving MSC therapy [27,28]. Therefore, unraveling the molecular pathways involved in MSC-induced immunosuppressive actions, including the generation of MDSCs and suppressive macrophages, is essential for designing safe and effective regimens for the treatment of immune-related diseases and cancers. Our study reveals the JNK pathway as one such mechanism mediating MSC-induced MDSC differentiation in the BM. As mimetics and inhibitors targeting the JNK pathway become available, their combination with MSC therapy holds promise for enhancing therapeutic efficacy and mitigating side effects in patients. 

In summary, our data demonstrate that MSCs activate the JNK–MAPK pathway in BM cells under GM-CSF stimulation to induce the expression of immunosuppressive molecules, promote MDSC differentiation and repress pro-inflammatory activation. 

## 4. Materials and Methods

### 4.1. Cell Culture 

The experimental protocol was approved by the Institutional Animal Care and Use Committee of the Seoul National University Biomedical Research Institute (Seoul, Republic of Korea). 

Single cell suspensions of BM cells were prepared by flushing the femur of 7-week-old C57BL/6 mice (Orient Bio, Seongnam, Republic of Korea). The cells were then filtered through a 70 μM cell strainer (#352350, Corning incorporation, Corning, NY, USA) and collected via centrifugation. After red blood cell lysis, the resulting cells were cultured in RPMI1640 media (Welgene, Daegu, Republic of Korea) with 10% (vol/vol) heat-inactivated fetal bovine serum (FBS) (Gibco, Waltham, MA, USA) and 1% penicillin-streptomycin (PS) (Lonza, Walkersville, MD, USA) at 37 °C in 5% CO_2_. For differentiation induction, the cells were treated with GM-CSF (40 ng/mL, GenScript, Piscataway, NJ, USA) for 5 d. 

In the LPS stimulation assay, GM-CSF-treated cells were exposed to 100 ng/mL LPS (InvivoGen, San Diego, CA, USA) for 18 h. 

For JNK inhibition, SP600125 (0, 10, 20 and 40 μM) (#tlrl-sp60, InvivoGen) was added to BM cells for 3 d. 

Human BM-derived MSCs were provided by the Center for the Preparation and Distribution of Adult Stem Cells, which supplied the standardized preparations of MSCs enriched for early progenitor cells to over 300 laboratories under the auspices of an NIH/NCRR grant (P40 RR 17447-06) and ethics approval. The cells consistently differentiated into three lineages in culture and were negative for hematopoietic markers (CD34 and CD45) and positive for CD44, CD73, CD90 and CD105, as examined via flow cytometry and single-cell RNA sequencing [16,29]. MSCs were cultured according to the instructions. Briefly, passage 2 MSCs were cultured at 37 °C in 5% CO_2_ until reaching 70–80% confluency (approximately 7 d). The culture medium consisted of α-minimal essential medium (Gibco), 17% FBS (Gibco), 1% PS (Lonza) and 2 mM L-glutamine (Lonza) and was changed every two days during culture. MSCs were harvested by incubating in 0.25% trypsin/1 mM ethylenediaminetetraacetic acid (EDTA) at 37 °C for 2 min and cocultured with BM cells at a ratio of 1:5 (MSCs:BM cells) in the BM cell culture media.

### 4.2. RNA Sequencing 

CD11b^hi^Ly6C^hi^ monocytes were isolated from GM-CSF-stimulated, MSC-uncocultured BM cells. CD11b^mid^Ly6C^mid^ MDSCs were sorted from GM-CSF-stimulated, MSC-cocultured BM cells. For FACS sorting, the BM cells were stained with mouse-specific anti-CD11b (clone M1/70, eBioscience, San Diego, CA, USA) and anti-Ly6C (clone HK1.4, eBioscience) antibodies (Abs) and sorted using a BD FACSAria™ III cell sorter (BD Biosciences, San Jose, CA, USA). 

For RNA-seq with the sorted BM cells, 500 ng of total RNA was extracted, and library construction was performed with the QuantSeq 3′ mRNA-Seq Library Prep Kit (Lexogen, Vienna, Austria). High-throughput sequencing was conducted as single-end 75 sequencing using NextSeq 500 (Illumina, San Diego, CA, USA). QuantSeq 3′ mRNA-Seq reads were aligned using Bowtie2, and DEGs were determined based on counts from unique and multiple alignments using coverage in Bedtools. The data were deposited at ArrayExpress accession E-MTAB-8975.

The up- or downregulated genes by more than 2-fold in CD11b^mid^Ly6C^mid^ cells relative to CD11b^hi^Ly6C^hi^ cells were selected and subjected to pathway analysis based on the KEGG database using DAVID Bioinformatics [14,15] 

### 4.3. Western Blot Assays

BM cells were lysed in a RIPA Buffer (Biosesang, Seongnam, Republic of Korea) including a protease inhibitor and phosphatase inhibitor (Thermo Fisher Scientific, Waltham, MA, USA), and they were sonicated on ice. After centrifugation, clear cell lysates were acquired and assayed for protein concentration. Following denaturation, 40 µg of protein was separated on a 4–12% SDS-PAGE gel (Invitrogen, Waltham, MA, USA) at 150 V for 90 min and transferred to a methanol-soaked PVDF membrane (Invitrogen) at 40 V for 100 min. The membranes were blocked with 3% bovine serum albumin for 1 h and then incubated at 4 °C overnight with anti-phopho-SAPK/JNK (Thr183/Tyr185) (#9255, Cell Signaling Technology, Danvers, MA, USA), anti-phopho-p38 MAPK (Thr180/Tyr185) (#9216, Cell Signaling Technology), anti-SAPK/JNK (#9252, Cell Signaling Technology) and anti-p38 MAPK (#9212, Cell Signaling Technology) Abs. The membranes were then incubated with anti-mouse or anti-rabbit IgG Abs conjugated to horseradish peroxidase (HRP) at room temperature for 1 h. After stripping at 55 °C for 30 min, they were incubated at room temperature for 1 h with anti-β-actin Abs (Santa Cruz Biotechnology, Dallas, TX, USA) and subsequently with anti-mouse HRP Abs (Cell Signaling Technology). The protein expression levels were normalized to the corresponding β-actin levels.

### 4.4. Quantitative Real-Time Reverse-Transcription PCR (qRT-PCR)

BM cells were lysed in an RNA isolation reagent (RNA Bee, Tel-Test, Friendswood, TX, USA) and homogenized with an ultrasound sonicator (Ultrasonic Processor, Cole Parmer Instruments, Vernon Hills, IL, USA). Total RNA was extracted using an RNeasy Mini kit (Qiagen, Hilden, Germany), and double-stranded cDNA was synthesized via reverse transcription (High Capacity RNA-to-cDNA^TM^ Kit, Applied Biosystems, Carlsbad, CA, USA). Real-time PCR amplification was performed using TaqMan^®^ Universal PCR Master Mix (Applied Biosystems, Carlsbad, CA, USA) in an automated instrument (ABI 7500 Real Time PCR System, Applied Biosystems, Carlsbad, CA, USA). Data were normalized to *Gapdh* and expressed as fold changes relative to the controls. All PCR probe and primer sets were mouse-specific (TaqMan^®^ Gene Expression Assay, Applied Biosystems, Carlsbad, CA, USA).

### 4.5. Enzyme-Linked Immunosorbent Assay (ELISA)

Cell-free supernatants were collected via centrifugation and analyzed for concentrations of mouse-specific IL-10, active TGF-β1 and active TGF-β2 (DuoSet^®^ ELISA, R&D Systems, Minneapolis, MN, USA) according to the manufacturer’s instructions. 

### 4.6. Statistical Analysis

After a normality test, data were analyzed either via one-way ANOVA followed by Tukey’s Honestly Significant Difference test or the Kruskal–Wallis test followed by Dunn’s multiple comparison test (GraphPad Prism Software 10.1.2, San Diego, CA, USA). Data were presented as the mean ± SD, and differences were considered significant at *p* < 0.05.

## Figures and Tables

**Figure 1 ijms-25-01119-f001:**
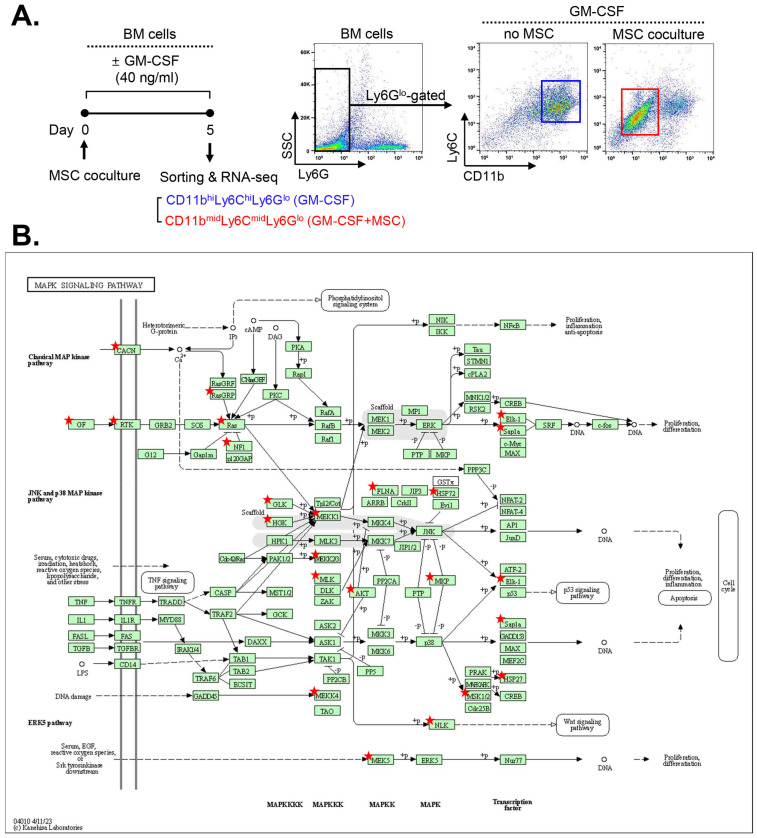
RNA-seq reveals upregulation of MAPK pathway-related genes in MSC-induced MDSCs. (**A**) Experimental design. CD11b^hi^Ly6C^hi^Ly6G^lo^ cells were isolated from BM cells cultured for 5 d under GM-CSF stimulation (40 ng/mL) without MSC coculture. CD11b^mid^Ly6C^mid^Ly6G^lo^ cells were sorted from GM-CSF-stimulated BM cells with MSC coculture. Both cell populations were comparatively analyzed via RNA-seq. (**B**) The KEGG pathway analysis using the DAVID tool. The MAPK signaling pathway included the highest number of genes upregulated in MSC-induced MDSCs (marked with red stars).

**Figure 2 ijms-25-01119-f002:**
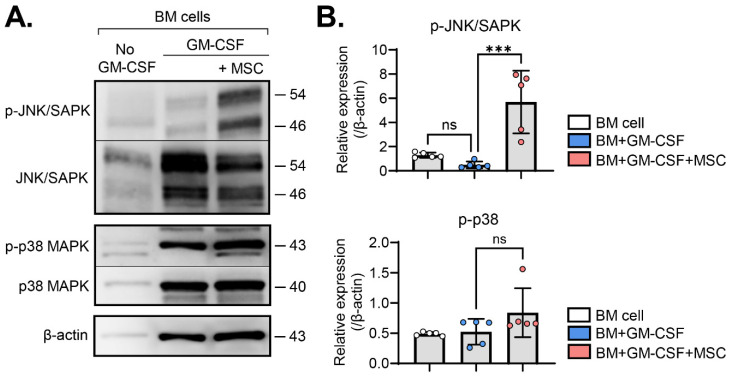
MSCs induce JNK activation in GM-CSF-stimulated BM cells. (**A**) Representative Western blot images. BM cells were subjected to Western blot analysis to assess total and phosphorylated forms of JNK/SAPK and p38 proteins under the indicated treatments. (**B**) Densitometric analysis of Western blot images. The relative amounts of phosphorylated forms of specified proteins were quantitated as a ratio of each protein band relative to β-actin. Mean values ± SD are shown. *** *p* < 0.001; ns: not significant via one-way ANOVA with Tukey’s multiple comparison test (p-JNK/SAPK) or Kruskal–Wallis test followed by Dunn’s multiple comparisons test (p-p38).

**Figure 3 ijms-25-01119-f003:**
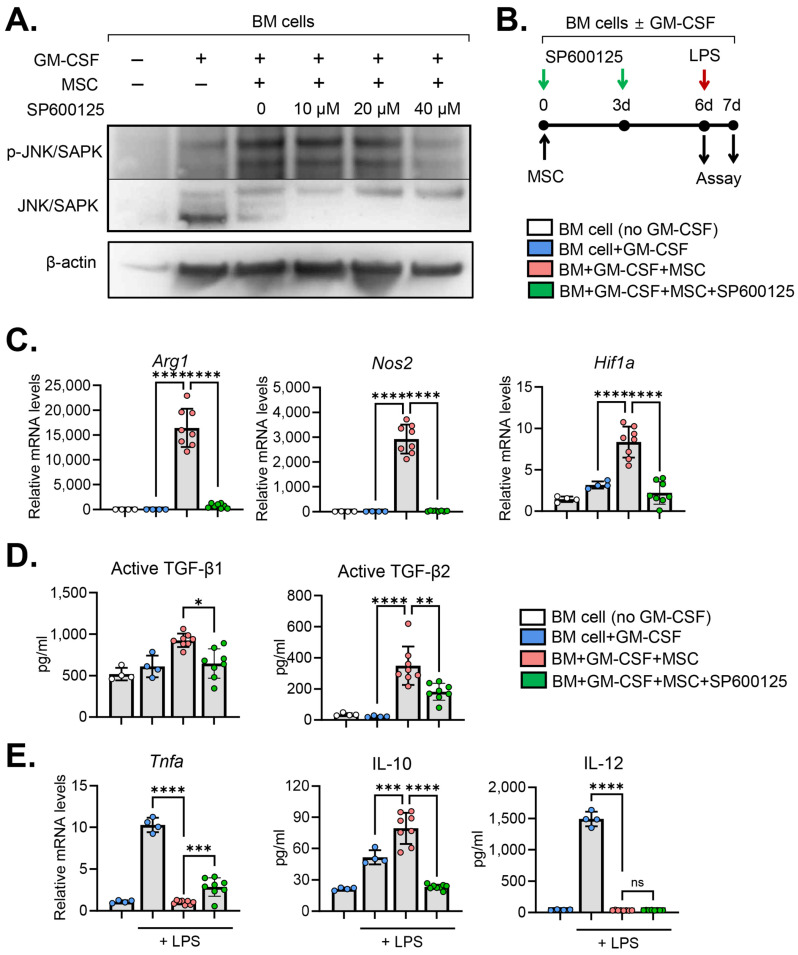
JNK inhibition reverses MSC effects on immunoregulatory molecules in BM cells. (**A**) Representative Western blot images. BM cells were subjected to Western blot analysis for total and phosphorylated forms of JNK/SAPK in BM cells under the indicated treatments. The cells were treated with incremental doses (10~40 μM) of SP600125, a JNK inhibitor, for 3 d. (**B**) Experimental scheme. For JNK inhibition, GM-CSF-stimulated BM cells were treated with 40 μM SP600125 every 3 d. For LPS stimulation assay, the cells were exposed to 100 ng/mL LPS for 18 h. (**C**) qRT-qPCR for MDSC hallmark and immunoregulatory genes *Arg1, Nos2* and *Hif1a* in BM cells. The mRNA levels are presented as fold changes relative to the levels in GM-CSF-untreated and MSC-uncocultured cells. (**D**) ELISA for secreted levels of immunoregulatory cytokines TFG-β1 and TGF-β2 (both active forms) in the supernatant of BM cell culture. (**E**) qRT-qPCR for *Tnfa* and ELISA for IL-10 and IL-12 in BM cells in response to LPS stimulation. Mean values ± SD are shown. * *p* < 0.05, ** *p* < 0.01, *** *p* < 0.001, **** *p* < 0.0001; ns: not significant via one-way ANOVA with Tukey’s multiple comparison test or Kruskal–Wallis test followed by Dunn’s multiple comparison test (active TFG-β1).

**Table 1 ijms-25-01119-t001:** Genes related to the MAPK pathway that were upregulated in MSC-induced MDSCs relative to pro-inflammatory monocytes derived from BM cells without MSC coculture.

Gene Description	Gene Symbol	Fold Change *
filamin, beta	*Flnb*	559.322
FMS-like tyrosine kinase 1	*Flt1*	223.11
neurofibromin 1	*Nf1*	179.681
ephrin A2	*Efna2*	135.17
calcium channel, voltage-dependent, P/Q type, alpha 1A subunit	*Cacna1a*	123.802
dual-specificity phosphatase 10	*Dusp10*	119.544
thymoma viral proto-oncogene 3	*Akt3*	112.279
mitogen-activated protein kinase kinase kinase 9	*Map3k9*	69.346
platelet-derived growth factor receptor, beta polypeptide	*Pdgfc*	37.249
fibroblast growth factor receptor 1	*Fgfr1*	29.61
calcium channel, voltage-dependent, N type, alpha 1B subunit	*Cacna1b*	27.199
related RAS viral (r-ras) oncogene 2	*Rras2*	21.369
mitogen-activated protein kinase kinase kinase 4	*Map3k4*	21.317
mitogen-activated protein kinase kinase kinase kinase 3	*Map4k3*	12.556
mitogen-activated protein kinase kinase kinase kinase 4	*Map4k4*	7.147
ELK1, member of ETS oncogene family	*Elk1*	12.104
nemo-like kinase	*Nlk*	6.99
calcium channel, voltage-dependent, L type, alpha 1D subunit	*Cacna1d*	5.488
hepatocyte growth factor	*Hgf*	3.193
mitogen-activated protein kinase kinase kinase 1	*Map3k1*	3.146
platelet-derived growth factor receptor, beta polypeptide	*Pdgfrb*	3.010
heat shock protein 1	*Hspb1*	2.873
mitogen-activated protein kinase kinase 5	*Map2k5*	2.75
heat shock protein 1B	*Hspa1b*	2.626
RAS, guanyl-releasing protein 3	*Rasgrp3*	2.563
fibroblast growth factor 7	*Fgf7*	2.471
mitogen-activated protein kinase kinase kinase 3	*Map3k3*	2.372
ribosomal protein S6 kinase, polypeptide 5	*Rps6ka5*	2.286
dual-specificity phosphatase 4	*Dusp4*	2.278
ELK4, member of ETS oncogene family (Elk4)	*Elk4*	2.264
dual-specificity phosphatase 8	*Dusp8*	2.192
fibroblast growth factor receptor 2	*Fgfr2*	2.189
nerve growth factor	*Ngf*	2.1

* The fold change values of upregulated genes in CD11b^mid^Ly6C^mid^Ly6G^lo^ MDSCs sorted from MSC-cocultured BM cells under GM-CSF stimulation relative to those in CD11b^hi^Ly6C^hi^Ly6G^lo^ monocytes sorted from GM-CSF-stimulated BM cells without MSC coculture.

**Table 2 ijms-25-01119-t002:** Genes related to the NOD-like receptor pathway that were downregulated in MSC-induced MDSCs relative to pro-inflammatory monocytes derived from BM cells without MSC coculture.

Gene Description	Gene Symbol	Fold Change *
toll-like receptor 4	*Tlr4*	0.009
thioredoxin 2	*Txn2*	0.012
BCL2-like 1	*Bcl2l1*	0.013
C-C motif chemokine ligand 5	*Ccl5*	0.014
gasdermin D	*Gsdmd*	0.014
interleukin 1 beta	*Il1b*	0.014
NLR family, apoptosis inhibitory protein 5	*Naip5*	0.016
v-rel reticuloendotheliosis viral oncogene homolog A	*Rela*	0.016
caspase 8	*Casp8*	0.018
C-X-C motif chemokine ligand 3	*Cxcl3*	0.02
mitochondrial antiviral-signaling protein	*Mavs*	0.021
nucleotide-binding oligomerization domain-containing 1	*Nod1*	0.023
TNF receptor-associated factor 6	*Traf6*	0.024
autophagy related 12	*Atg12*	0.032
signal transducer and activator of transcription 1	*Stat1*	0.032
NLR family, apoptosis inhibitory protein 2	*Naip2*	0.039
inhibitor of kappaB kinase beta	*Ikbkb*	0.044
TGF-beta-activated kinase 1/MAP3K7 binding protein 1	*Tab1*	0.048
TNF receptor-associated factor 3	*Traf3*	0.051
interleukin 18	*Il18*	0.053
inositol 1,4,5-triphosphate receptor 2	*Itpr2*	0.054
dynamin 1-like	*Dnm1l*	0.055
tumor necrosis factor	*Tnf*	0.06
RanBP-type and C3HC4-type zinc finger containing 1	*Rbck1*	0.074
nuclear factor of kappa light polypeptide gene enhancer in B cells 1, p105	*Nfkb1*	0.085
thioredoxin 1	*Txn1*	0.086
transient receptor potential cation channel, subfamily M, member 2	*Trpm2*	0.088
cathepsin B	*Ctsb*	0.115
inositol 1,4,5-trisphosphate receptor 1	*Itpr1*	0.117
interferon-activated gene 204	*Ifi204*	0.138
C-X-C motif chemokine ligand 1	*Cxcl1*	0.14
GABA type A receptor-associated protein-like 1	*Gabarapl1*	0.142
NLR family, apoptosis inhibitory protein 6	*Naip6*	0.143
inhibitor of kappaB kinase epsilon	*Ikbke*	0.153
C-X-C motif chemokine ligand 2	*Cxcl2*	0.167
interferon regulatory factor 3	*Irf3*	0.173
tumor necrosis factor, alpha-induced protein 3	*Tnfaip3*	0.179
voltage-dependent anion channel 3	*Vdac3*	0.194
DEAH-box helicase 33	*Dhx33*	0.224
nuclear factor of kappa light polypeptide gene enhancer in B cells inhibitor, alpha	*Nfkbia*	0.308
GABA type A receptor-associated protein-like 2	*Gabarapl2*	0.423

* The fold change values of downregulated genes in CD11b^mid^Ly6C^mid^Ly6G^lo^ MDSCs derived from MSC-cocultured BM cells under GM-CSF stimulation relative to those in CD11b^hi^Ly6C^hi^Ly6G^lo^ monocytes sorted from GM-CSF-stimulated BM cells without MSC coculture.

## Data Availability

The RNA-seq data have been deposited in ArrayExpress under Accession code E-MTAB-8975. Other data presented in this study are available on request from the corresponding author.

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
