# Peer review of "Mesenchymal Stem/Stromal Cells Induce Myeloid-Derived Suppressor Cells in the Bone Marrow via the Activation of the c-Jun N-Terminal Kinase Signaling Pathway"

_ijms, 2024, doi:10.3390/ijms25021119_

Round 1

Reviewer 1 Report

Comments and Suggestions for Authors

The study is relevant in order to investigate signaling pathways and interaction mechanisms between cells through mainly variation in gene expression. However, the study could be better organized and described more clearly. Some considerations:

Introduction: Better characterize MSCs and myeloid-derived suppressor cells, highlighting their biological properties and the relationship between these cells, as well as the relationship between these cells and defense cells/biomolecules of the inflammatory process. Further detail the role of MSCs and myeloid-derived suppressor cells in regulating immunomodulation and inflammation, adding more studies.

Materials and Methods: The methodology could be better detailed, in order to allow the reproducibility of the study, especially in the sections referring to cell culture and RNA sequencing (highlight the adopted fold-change). Add information about the ethical approval protocol for the use of Human BM-derived MSCs.

Results: The genes in Table 1 could be organized in descending order according to fold-change. Gene symbols should be in italics. In order to provide an overview of gene expression variations (upregulated genes and downregulated genes), the study must contain a Table with the main downregulated genes.

Discussion: Add a discussion about expression variation in target genes, including up- and down-regulated genes, as well as the main biological processes related to these genes, in order to provide an overview of the study findings, relating them with literature. Add more studies, if applicable. This complementation may better support the Authors' conclusion, elucidating in more detail the mechanisms and signaling pathways investigated.

Author Response

We appreciate the reviewer for the useful and constructive comments. They were certainly helpful for improving our paper. We have revised the manuscript based on the comments. Changes to the text in the manuscript are blue-colored. The following is our point-by-point response to the comments.

  1. Introduction: Better characterize MSCs and myeloid-derived suppressor cells, highlighting their biological properties and the relationship between these cells, as well as the relationship between these cells and defense cells/biomolecules of the inflammatory process. Further detail the role of MSCs and myeloid-derived suppressor cells in regulating immunomodulation and inflammation, adding more studies.
  • Response: Per the reviewer’s comment, we have supplemented the Introduction with more detailed information and references (Line 34-55, new Ref 3-7).

  1. Materials and Methods: The methodology could be better detailed, in order to allow the reproducibility of the study, especially in the sections referring to cell culture and RNA sequencing (highlight the adopted fold-change). Add information about the ethical approval protocol for the use of Human BM-derived MSCs.
  • Response: Per the reviewer’s comment, we have added the information to the Methods section (Line 319-326, 348-350, new Ref 14-16, 29).

  1. Results: The genes in Table 1 could be organized in descending order according to fold-change. Gene symbols should be in italics. In order to provide an overview of gene expression variations (upregulated genes and downregulated genes), the study must contain a Table with the main downregulated genes.
  • Response: As suggested, we have reorganized Table 1 in descending order of fold change (new Table 1). Also, we have added the data on the pathways with the downregulated genes to the revised manuscript (Line 149-155, Table 2).

  1. Discussion: Add a discussion about expression variation in target genes, including up- and down-regulated genes, as well as the main biological processes related to these genes, in order to provide an overview of the study findings, relating them with literature. Add more studies, if applicable. This complementation may better support the Authors' conclusion, elucidating in more detail the mechanisms and signaling pathways investigated.
  • Response: As suggested, we have supplemented the Discussion with more detailed information (Line 263-271).

Reviewer 2 Report

Comments and Suggestions for Authors

This is a straight forward manuscript/research project that clearly demonstrates the role of the JNK signalling pathway in MSC indiced MDSC activation. It is well known that one of the therapeutic properties of MSC is the ability to modulate the immune system. For many diseases/injuries the ability of MSC to suppress inflammation for example is documented. What is not as clear is the mechanism. This manuscript reports on the role of JNK signalling through a series of experiments that include the use of specific inhibitors to identify which pathways are involved and which are not.

Importantly the authors, in the discussion, bring up the important point that MSC interactions are complex and this study defines one of many possible biological pathways/mechanisms that are involved in MSCs role in immune modulation.

The manuscript is well written, the experiments are clearly documented with controls and the figures are easy to follow.

Author Response

The authors are grateful for insightful and encouraging comments the reviewer has given. Thank you for your time and effort!

Round 2

Reviewer 1 Report

Comments and Suggestions for Authors

The adjustments improved the manuscript.